# Safety Assessment of a Sublingual Vaccine Formulated with Poly(I:C) Adjuvant and Influenza HA Antigen in Mice and Macaque Monkeys: Comparison with Intranasal Vaccine

**DOI:** 10.3390/vaccines13030261

**Published:** 2025-02-28

**Authors:** Tetsuro Yamamoto, Fusako Mitsunaga, Atsushi Kotani, Kazuki Tajima, Kunihiko Wasaki, Shin Nakamura

**Affiliations:** 1Innovation Research Center, EPS Holdings, Inc., 2-1 Tsukudohachimancho, Shinjuku-ku, Tokyo 162-0815, Japan; yamamoto.tetsuro061@eps.co.jp (T.Y.); kotani.atsushi332@eps.co.jp (A.K.); tajima.kazuki374@eps.co.jp (K.T.); wasaki.kunihiko377@eps.co.jp (K.W.); 2EP Mediate Co., Ltd., 1-8 Tsukudocho, Shinjuku-ku, Tokyo 162-0821, Japan; 3Research Center, EPS Innovative Medicine Co., Ltd., 1-8 Tsukudocho, Shinjuku-ku, Tokyo 162-0821, Japan; 4Intelligence & Technology Lab, Inc., 52-1 Fukue, Kaizu-cho, Kaizu 503-0628, Japan; mitsunaga@itechlab.co.jp; 5Biomedical Institute, NPO Primate Agora, 52-2 Fukue, Kaizu-cho, Kaizu 503-0628, Japan

**Keywords:** administration route, olfactory sensitivity, TNF gene, Bell’s palsy, brain

## Abstract

A sublingual vaccine comprising the Poly(I:C) adjuvant and influenza HA antigen was evaluated for safety in both mice and macaque monkeys relative to its intranasal counterpart. Safety was assessed in terms of harmful effects corresponding to the upregulation of the inflammation-associated genes *Saa3*, *Tnf*, *IL6*, *IL1b*, *Ccl2*, *Timp1*, *C2*, *Ifi47*, *Aif1*, *Omp*, *Nos2*, and/or *Gzmb* in mice and *SAA2*, *TNF*, *IL6*, *IL1B*, *CCL2*, *TIMP*, *C2*, *AIF1*, and *GZMB* in macaques. Quantitative gene expression analyses were performed using RT-qPCR with RNA samples from four tissue types, the olfactory bulb, pons, lung, tongue, and lymph node, from both mice and macaques. In mice, the intranasally delivered vaccine markedly upregulated the inflammation-related genes in the olfactory bulb 1 day and 7 days after vaccination. The adverse effects of intranasal vaccination were also observed in macaques, albeit to a lesser extent than in mice. The intranasal vaccination also upregulated these genes in the pons of both mice and macaques. In contrast, the sublingual vaccine did not adversely affect the olfactory bulb or pons in either mice or macaques. The intranasally administered vaccine significantly upregulated these genes in the lungs only 1 day after vaccination, but not 7 days later, in both mice and macaques. We conclude that intranasal vaccination results in unfavorable side effects corresponding to upregulated inflammatory genes in the brain (olfactory bulb and pons). Sublingual vaccination, however, did not induce these side effects in either mice or macaques and was hence evaluated as safe.

## 1. Introduction

The oral and nasal cavities are the first regions of contact with all external non-self materials, including pathogenic microorganisms and allergic compounds. Airborne viruses such as influenza and severe acute respiratory syndrome coronavirus 2 (SARS-CoV-2) enter from these upper respiratory tract mucosae. Therefore, these mucosae function as the first line of defense against these viruses. The mucosal protection generally operates through both mucosal and systemic immune responses, resulting in antibody-mediated and cytotoxic T cell reactions. Sublingual or intranasal vaccines elicit these immune responses, eliminate the need for medical staff, and allow needle-free administration. Vaccines are primarily categorized into three types: RNA/DNA vaccines, protein-based vaccines, and live-attenuated/killed vaccines. During the COVID-19 pandemic, gene-based RNA or DNA vaccines were employed to combat the disease; however, these vaccines were associated with side effects, including fever, headache, nausea, and chills [1]. Conversely, protein-based vaccinations exhibit fewer adverse effects and are employed in the prevention of hepatitis and several viral illnesses [2]. Although the development of a protein vaccine for SARS-CoV-2 requires time, it is anticipated to become a cornerstone in the global protection against COVID-19 [3]. A protein-based vaccine requires an immunity-stimulating adjuvant, which are reviewed in a recent article [4]. Among them there are two interesting adjuvants. One is an oil-in-water nano-emulsion vaccine, MF59 or AS03, that stimulates Th1/Th2 cytokines [5]. The other is a double-stranded RNA (polyinosinic–polycytidylic acid) vaccine, Poly(I:C), which is a ligand for Toll-like receptor (TLR) 3 to induce immunological and proinflammatory responses [6]. MF59 and AS03 received approval as adjuvants for intramuscularly administered influenza vaccinations [7]. Nevertheless, Poly(I:C) remains unapproved because of its adverse effects, including fever and the generation of proinflammatory cytokines.

The vaccination site is also a limiting element in developing a protein-based vaccine. Protection against upper respiratory tract viruses, such as SARS-CoV-2 and influenza, can be provided using sublingual or intranasal vaccines. Sublingual vaccines, like nasal vaccines, elicit mucosal immune responses in the upper and lower respiratory tracts, stomach, small intestine, and reproductive tract, as well as a systemic response [8,9]. Another advantage of sublingual vaccination is that it is safer than intranasal vaccines, which can harm the brain, central nervous system, and lungs [10,11]. The sublingual vaccine is also administered needle-free, resulting in good patient compliance and the ability to self-administer without the assistance of medical personnel. However, administering vaccines sublingually has practical issues, such as the mucin barrier that inhibits vaccine access into immune cells and the large volume of saliva that dilutes the vaccine.

We previously developed a sublingual vaccine formulation with the Poly(I:C) adjuvant and SARS-CoV-2 receptor-binding domain (RBD) [12,13] or influenza HA [14] antigens using a non-human primate model, cynomolgus macaques. These sublingual Poly(I:C)-adjuvanted vaccines induced mucosal and systemic immunological responses, resulting in antigen-specific antibodies in the saliva, nasal washes, and blood. In these studies, we prevented mucin inhibition by pre-treating the sublingual surface with N-acetyl cysteine (NAC), a moderately reducing reagent that disintegrates the mucin layer [12,15,16]. We also settled saliva dilution by using an anesthetic, a combination of medetomidine and ketamine, to decrease the saliva output during vaccination [12]. DNA microarray analysis revealed that sublingual Poly(I:C)-adjuvanted vaccinations elicited immunological responses via a previously unknown mechanism that generates balanced activation and inhibition in a “Yin/Yang” concept [13,14]. The Poly(I:C)-adjuvanted vaccines administered via the sublingual route appeared to be safe based on gene expression analyses of proinflammatory-related factors in comparison with the AddaS03 adjuvant, which has the same composition as AS03 [13], but its safety remains to be further evaluated.

This study aimed to thoroughly evaluate the safety of sublingual Poly(I:C)-adjuvanted vaccinations. The evaluation was conducted through quantitative gene expression analyses (RT-qPCR) of inflammatory-related factors in several nervous system tissues from both mice/rodents and macaque monkeys/non-human primates that were administered a Poly(I:C)-adjuvanted influenza HA vaccine via the sublingual and intranasal routes. Intranasally administered Poly(I:C)-adjuvanted vaccination significantly upregulated the expression of inflammatory-related genes in the olfactory bulbs of mice, unlike the sublingually administered vaccine, which did not produce adverse effects in either mice or macaques. The harmful effect caused by the intranasally administered vaccination was also noted in the olfactory bulbs of macaques, although the effects were less pronounced. The previously reported adverse effects of the Poly(I:C)-adjuvanted vaccination resulted in particular events in the olfactory bulbs of rodents. Consequently, the sublingual Poly(I:C)-adjuvanted vaccination appears to be safe in primates, although its safety in humans remains to be studied clinically.

## 2. Materials and Methods

### 2.1. Reagents

NAC, bovine serum albumin, Na-Casein, and sodium azide (NaN_3_) were obtained from the FUJIFILM Wako Pure Chemical Corporation (Osaka, Japan). Phosphate-buffered saline (PBS; Nissui, Tokyo, Japan), a quadrivalent FLUBIK HA Syringes™ vaccine (The Research Foundation for Microbial Diseases of Osaka University, Suita, Japan), and Poly(I:C) HMW vaccine-grade (Invitrogen, Waltham, MA, USA) were also used. RNAiso Plus, PrimeScript™ Reverse Transcriptase, 2680, Recombinant RNase Inhibitor, TB Green^®^ Premix Ex Taq™ (Tli RNaseH Plus), RR420 (Takara Bio Inc., Kyoto, Japan), the RNeasy MinElute Cleanup Kit ((QIAGEN, Tokyo, Japan), dNTP Mix and Oligo(dT)15 Primer (Promega, Tokyo, Japan), the Low Input Quick Amp Labeling Kit, and the RNA6000 Nano Kit (Agilent Technologies, Santa Clara, CA, USA) were used in this study.

### 2.2. Animals

Thirty-two male mice (IRC; aged 7 to 8 weeks) and six male macaque monkeys (*Macaca fascicularis* and *Macaca mulatta*; aged 15 to 20 years) were used. In accordance with the 3R policy for animal use, the number of macaque monkeys was minimized. The monkeys tested negative for B virus, simian immunodeficiency virus, TB, *Shigella* spp., *Salmonella* spp., and helminthic parasites.

The animal examinations were performed in accordance with the regulations set forth by the Institutional Animal Care and Use Committee of the Intelligence and Technology Lab, Inc. (ITL; Kaidu, Japan), adhering to the standards for the Proper Conduct of Animal Experiments. The Animal Care Committee of the ITL sanctioned these examinations, designating them with the code AE2022022 and granting approval on 24 November 2022. The ITL Biosafety Committee has also sanctioned additional studies.

### 2.3. Vaccination and Sampling

The preparation and administration of a vaccine produced with the Poly(I:C) adjuvant and influenza HA antigen were performed as previously described [14]. The following procedures were used to administer the vaccines and collect samples from the mice and macaques (Figure 1).

#### 2.3.1. In Mice

Figure 1A shows an outline of the vaccination and sampling in mice. Thirty-two mice were divided into eight groups, A1 to A8, with each group including four animals. Groups A1, A2, A5, and A6 were designated for intranasal administration, whereas groups A3, A4, A7, and A8 were allocated to sublingual vaccination. Groups A1 and A5 were administered PBS as a control in the oral route, whereas groups A3 and A7 received PBS as a control in the sublingual route. Groups A2 and A6 received the Poly(I:C)-adjuvanted vaccine via the intranasal route, whereas groups A4 and A8 were vaccinated sublingually. The animals were given either 10 μL/head of PBS in the control groups (A1, A5, A3, and A7) or 10 μL/head of vaccine for each of the vaccinated groups (A2, A6, A4, and A8). Mice from groups A1 to A4 and A5 to A8 were euthanized to obtain blood and tissue samples at 1 day or 7 days post-administration of PBS or the vaccine, respectively. Tissue samples from the olfactory bulb, pons, lung, tongue, and (submandibular) lymph node were obtained at two time points.

#### 2.3.2. In Macaque Monkeys

Figure 1B shows six macaque monkeys divided into three groups, B1 to B3, with each group consisting of two macaques. Group B1 was used as a control for the intranasal or sublingual administration of PBS. Group B2 (intranasal route) and Group B3 (sublingual route) received the vaccine under the previously described conditions [14]. Briefly, macaques were administered either 500 μL of PBS per head in the control group (B1) or 500 μL of the vaccine per head in the vaccinated groups (B2 and B3). Macaques from three groups (B1, B2, and B3) were euthanized to collect blood and tissue 7 days post-administration of PBS or the vaccine. Tissue samples from the olfactory bulb, pons, lung, tongue, and (submandibular) lymph node were obtained at the indicated time point.

### 2.4. Blood Testing

In mice, fresh blood samples were collected at two time points, 1 and 7 days after vaccination via the intranasal or sublingual routes. After the centrifugation of the blood, the plasma samples were assayed for 13 biochemical tests via the colorimetric enzyme assay using a Hitachi Automatic Analyzer 3500: total protein, albumin, albumin/globulin ratio, total bilirubin, aspartate transaminase (glutamic oxaloacetic transaminase), alanine transaminase (glutamic pyruvic transaminase), alkaline phosphatase, gamma-glutamyl transpeptidase, urea nitrogen–BUN, creatinine, total cholesterol, neutral fats, and C-reactive protein (CRP).

In macaque monkeys, fresh blood samples were collected 7 days after sublingual or intranasal vaccination. Whole blood samples were examined by flow cytometry using Sysmex Nonclinical Blood Testing and Cell Analysis NX-31, with a complete blood count of eight items: red blood cells, white blood cells (WBC), hemoglobin, hematocrit, mean cell volume, mean corpuscular hemoglobin, mean corpuscular hemoglobin concentration, and platelets. Plasma samples were assayed for the same 13 items as in mice.

### 2.5. RNA Isolation

Tissue samples from each animal in the experimental groups, A1~A8 for mice and B1~B3 for macaques, were used for RNA preparation. RNA isolation and its quality tests were performed as previously described [13]. In both mice and macaques, equal amounts of purified RNA from each tissue of the individual animals were combined and pooled per group and then used for gene expression analyses.

### 2.6. Gene Expression Analyses Using Quantitative Reverse Transcription PCR (RT-qPCR)

The mRNA levels of target genes in the tissue samples were determined using RT-qPCR, as described previously [13]. cDNA was synthesized from the pooled RNA using Prime Script Reverse Transcriptase with RNase Inhibitor (Takara Bio Inc., Kyoto, Japan), a dNTP mixture (Promega Corp.), and Oligo dT primers (Invitrogen, Waltham, MA, USA). Real-time PCR was performed using the Mx3000P QPCR System (Agilent Technologies, Santa Clara, CA, USA) with an SYBR Premix Ex Taq II (Tli RNase H Plus) Kit (Takara Bio Inc., Kyoto, Japan). Specific primers for nine mouse genes, *Saa3*, *Tnf*, *Il1b*, *Il6*, *Ifi4*, *Ccl2*, *Timp1*, *C2*, and *Gzmb*; nine monkey genes, *SAA2*, *TNF*, *IL1Bb*, *IL6*, *AIF1*, *CCL2*, *TIMP1*, *C2*, and *GZMB*; and the reference gene low-density lipoprotein receptor-related protein 10 (Lrp10/LRP10) were designed using Primer3 and Primer-BLAST [17]. A standard curve was generated by the serial dilution of a known amount of glyceraldehyde 3-phosphate dehydrogenase amplicon to calculate the cDNA copy numbers of the genes. The PCR conditions included initial denaturation at 95 °C for 15 s, followed by 35 cycles of denaturation at 95 °C for 10 s and annealing/extension at 63 °C for 30 s, with a dissociation curve. The quantity of target gene mRNA was expressed as the ratio against that of a suitable reference gene, LRP10 [18]. The relative gene expression (/Control) indicates the fold change produced by vaccination in relation to the control gene expression on PBS administration.

### 2.7. Histological Examination

Tissue samples were collected from three macaque groups, B1 (control), B2 (intranasal vaccine), and B3 (sublingual vaccine), 7 days post-vaccination and subsequently fixed with formaldehyde. A paraffin block of the formaldehyde-fixed samples was sectioned into 4 μm slices using a microtome, REM-710 (Yamato Kohki Industrial, Asaka, Japan). The sections were stained with hematoxylin–eosin (HE) and subsequently examined using an Olympus BX43 optical microscope (Evident, Tokyo, Japan) under ×10 eye and ×10 and/or ×40 objective lenses.

## 3. Results

### 3.1. Blood Testing

Blood tests were conducted to evaluate the deleterious effects of a Poly(I:C)-adjuvanted sublingual and intranasal vaccine in both mice and macaque monkeys. Plasma samples from mice vaccinated sublingually and intranasally were analyzed in 13 biochemical blood tests (total protein, albumin, albumin/globulin ratio, total bilirubin, aspartate and alanine transaminases, alkaline phosphatase, gamma-glutamyl transpeptidase, urea nitrogen, creatinine, total cholesterol, neutral fats, and CRP) at 1 day and 7 days post-vaccination. Compared to the control group, the 13 biochemical parameters exhibited minimal variation in both sublingually and intranasally vaccinated mice, whereas modest individual differences were observed.

In macaque monkeys, fresh blood samples were analyzed for complete blood counts of eight parameters (red blood cells, WBCs, hemoglobin, hematocrit, mean cell volume, mean corpuscular hemoglobin, mean corpuscular hemoglobin concentration, and platelets) 7 days post-sublingual and intranasal vaccination. Plasma samples were analyzed for the same 13 biochemical parameters as in mice at 7 days post-sublingual and intranasal vaccination. Minimal differences were observed between the control group and the vaccinated monkeys in the complete blood counts and biochemical blood tests, except for individual variations. The levels of CRP in the plasma rose by 1.5 to 2.5 times in the sublingual and intranasal groups one day after vaccination. They also rose by 2.5 times in the control/PBS group. These results showed that the rise in plasma CRP was probably more due to the stress of the experiment than to the vaccine itself.

Thus, the Poly(I:C)-adjuvanted vaccine, administered sublingually or intranasally to both mice and monkeys, had negligible adverse effects on blood tests conducted 7 days post-vaccination.

### 3.2. Gene Expression Analyses of Inflammation-Related Genes

To assess the deleterious effects of the Poly(I:C)-adjuvanted sublingual or intranasal vaccines in both mice and macaque monkeys, quantitative gene expression analyses using RT-qPCR were conducted for inflammation-related genes, utilizing RNA from several tissue types. Table 1 and Table 2 represent the target genes, tissue samples, and sampling time points for the gene expression analysis.

#### 3.2.1. In Mice

Table 1 shows the selection of inflammation-associated genes in mice, comprising eight common genes (*Saa3*, *Tnf*, *IL-6*, *IL-1b*, *Ccl2*, *Timp1*, *C2*, and *Ifi4*) with additional genes (*Aif1*, *Omp*, *Nos2*, and *Gzmb*) for gene expression analysis. RNA samples from five tissue types—the olfactory bulb, pons, lung, tongue, and lymph node—which were obtained 1 day and 7 days post-vaccination, respectively, were used (Table 1). Figure 2 shows the changes in the expression of the inflammation-associated genes in the five tissue types at 1 day post-vaccination with the intranasal vaccine. In the olfactory bulb, the remarkable upregulation of two genes, *Saa3* and *Tnf*, was observed at rates of 3.8 to 5.8 times, respectively, alongside the significant upregulation of additional genes ranging from 1.7 to 2.8 times. In the pons, moderate upregulation of 1.5 to 2.3 times in the expression of all 10 genes was observed. In the lungs, the huge upregulation of five genes—*Saa3*, *Tnf*, *IL-6*, *Ccl2*, and *Timp1*—was observed, ranging from 5.5 to 13.5 times. Conversely, the sublingual vaccine exhibited little effect on the expression profile of inflammation-related genes in the olfactory bulb and other tissue types at 1 day post-vaccination (Figure 2). Figure 3 displays the effects of the intranasal vaccine on the expression of these genes at 7 days post-vaccination. The intranasal vaccine upregulated inflammatory-related genes in both the olfactory bulb and pons at 7 days post-vaccination, whereas the sublingual vaccine had negligible effects on the upregulation of these genes in the four tissue types, except for the lymph node, where the slight upregulation of a few genes was observed (Figure 3).

In mice, the intranasal administration of the Poly(I:C)-adjuvanted vaccine upregulated the inflammation-related genes in the olfactory bulb and pons at both 1 day and 7 days post-vaccination, as well as in the lungs at 1 day post-vaccination. On the other hand, the sublingual administration of the Poly(I:C)-adjuvanted vaccine resulted in little upregulation of these genes in the examined tissues after both 1 and 7 days. These results indicate that the Poly(I:C)-adjuvanted vaccination exhibits negligible side effects when administered sublingually in mice.

#### 3.2.2. In Macaque Monkeys

Gene expression analyses of macaque monkey samples were performed to complement the above-mentioned results in mice because of the differences in nasal structure and function between rodents and primates, including humans. Tissue samples from macaques were obtained from the smallest number of animals, specifically two individuals for each of the control, sublingual, and intranasal vaccination groups. Nine inflammation-associated genes in macaques—*SAA2*, *TNF*, *IL6*, *IL1B*, *CCL2*, *TIMP*, *C2*, *AIF1*, and *GZMB*—were chosen for RT-qPCR analysis, utilizing RNA from five tissue types, the olfactory bulb, pons, lung, tongue, and lymph node, collected 7 days post-vaccination (Table 2). Figure 4 shows that intranasal vaccination significantly upregulated the expression of five genes—*TNF*, *IL6*, *IL1B*, *TIMP1*, and *AIF1*—1.7- to 2.8-fold in the olfactory bulb at 7 days post-immunization, but sublingual vaccination resulted in little upregulation of these genes. The intranasal vaccine led to the moderate upregulation, approximately 1.4- to 1.8-fold, of numerous genes, including *SAA2*, *TNF*, *C2*, and *GZMB*, in the pons (Figure 4). The sublingual vaccination elicited a slight increase, approximately 1.2- to 1.4-fold, in the expression of several genes, specifically *SAA2*, *IL6*, and *CCL2*, in the lymph nodes at 7 days post-vaccination (Figure 4). No upregulation of gene expression was observed in either the lungs or tongue at 7 days post-sublingual and/or intranasal immunization (Figure 4).

In macaques, the intranasally administered vaccines also resulted in minimal unfavorable effects in both the olfactory bulb and pons, but sublingual vaccination had few side effects in these tissue types. Sublingual vaccination elicited the slight upregulation of these genes in the lymph nodes, indicating the activation of an immune–inflammatory response in the lymphatic sites near the vaccination. The gene expression analyses of inflammation-associated factors suggest that the sublingual administration of the Poly(I:C)-adjuvanted vaccine is safe, especially for macaque monkeys.

### 3.3. Histological Examination

Histological examinations were performed on macaque monkey samples because the structure and/or function of the oral and nasal cavities, other biological characteristics, and/or the immune–inflammatory responses are quite similar to those of humans. Patho-histopathological investigations were conducted using optical microscopy on HE-stained tissue specimens taken from macaques who were given PBS (control) or the Poly(I:C)-adjuvanted vaccine via the sublingual and intranasal routes, respectively. Figure 5 shows typical histological photographs for the intranasally vaccinated group, specifically the olfactory bulb, pons, and lungs (Figure 5A), and the sublingually vaccinated group, specifically the tongue and lymph nodes (Figure 5B). Intranasally vaccinated macaques had trace amounts of infiltrated white cells, lymphocytes, macrophages, and eosinophils in the olfactory bulb, pons, and lungs (Figure 5A), but the same amount of infiltrated white cells was detected in tissue specimens from the control animals. The slight infiltration of white cells was found in both the tongue and the lymph nodes (Figure 5B), as seen in the control animal samples.

Thus, Poly(I:C)-adjuvanted vaccination appears to have no detrimental effects related to the vaccinated site, namely the sublingual or intranasal cavities, in macaques.

## 4. Discussions

### 4.1. Poly(I:C) Adjuvant

An immunity-stimulating adjuvant is a crucial component for the development of a protein-based vaccine. Pathogen-associated molecular patterns (PAMPs) are adjuvants for numerous subunit vaccines. Poly(I:C) serves as a PAMP adjuvant to stimulate several components of the host defense in a manner analogous to a viral infection. Poly(I:C) is a synthetic double-stranded RNA molecule that activates both innate and adaptive immune responses. It mimics viral infections and elicits host immune responses by activating particular pattern recognition receptors (PRRs). Poly(I:C) primarily signals through TLR3, a transmembrane and mostly endosomal receptor, and MDA-5, a cytoplasmic receptor. Poly(I:C) induces an interferon response through TLR3 and MDA-5 signaling, resulting in the production of proinflammatory cytokines, chemokines, and/or costimulatory factors [6]. Poly(I:C) remains unapproved owing to its adverse effects, including fever and the production of proinflammatory cytokines. This is why more preclinical investigations on its safety utilizing non-human primates (NHPs) are needed. In previous studies, we developed a Poly(I:C)-adjuvanted vaccine administered via the sublingual cavity [12]. The sublingual vaccination adjuvanted with Poly(I:C) appeared to be safe based on preclinical examinations [13,14]. The present work thoroughly evaluated the safety of the Poly(I:C)-adjuvanted vaccine by comparing its sublingual and intranasal administration in both mice/rodents and macaques/NHPs, as discussed below.

### 4.2. Safety Assessment of the Vaccine in Mice

The administration site or route is another limiting aspect in the development of effective vaccinations. Airborne viruses, such as SARS-CoV-2 and influenza, invade the mucosal surfaces of the upper respiratory tract. Consequently, the vaccine ought to induce the production of virus antigen-specific IgA in the mucosal tissue of the oral cavity and/or nasal passages, rather than generating specific IgG or IgA in the blood [7]. Recently, alternate administration routes, such as oral delivery, for allergy immunotherapy or vaccination have been developed to provoke mucosal immune responses distinct from systemic responses [8]. Vaccinations via these approaches have greater efficacy than conventional subcutaneous vaccinations. Despite the establishment and partial practical application of intranasal vaccines [9], adverse effects on the brain/central nervous system or lungs have been described for intranasal administration [10,11,20]. In contrast, the oral/sublingual vaccine has satisfactory efficacy and improved safety, lacking adverse effects on the brain [28]. In primates, including monkeys and humans, the sublingual region possesses ample space, making it more conducive to vaccination compared to the nasal cavity. Sublingual vaccination, therefore, should better activate the mucosal immune response. Moreover, the previously mentioned adverse effects of the Poly(I:C) adjuvant were reported in intranasal administration using a mouse model [11,19,20]. The reported side effects probably resulted from differences in adjuvant reactivity between rodents and primates, related to the nasal structure and function [29], genomic characteristics, and the immune response [30]. Notably, Poly(I:C) serves as the most effective inducer of type I interferons among TLR agonists, activating the proinflammatory cytokine pathway in rodents [31].

Mice/rodents exhibit nocturnal behavior and possess an exceptionally acute sense of smell, facilitated by their olfactory system and associated neural pathways, enabling their survival in darkness. The olfactory system of mice exhibits higher reactivity to foreign compounds, including vaccines that are administered through the intranasal route. Conversely, macaques/primates are diurnal and do not require sensitive olfactory reactivity as in mice/rodents. These biomedical factors explain why the intranasally administered vaccination elicited significant adverse effects in mice but not in macaques.

In preclinical evaluations of vaccinations and/or medications administered intranasally in mice/rodents, it is necessary to account for the hypersensitive reactivity of the nasal and olfactory systems, which differ from those in macaques/primates.

### 4.3. Gene Expression Analyses for Vaccine Safety Evaluation

Gene expression analysis is a method for the precise monitoring of intracellular molecular events at the mRNA level, serving as precursors for proteins, enzymes, cytokines, and inflammatory mediators. Real-time RT-PCR (RT-qPCR) is a reliable technique to quantify changes in mRNA expression. RT-qPCR assays of proinflammatory genes have been suggested as a valuable tool for the safety assessment of vaccinations [32,33]. 

In our prior preclinical studies using macaques, we reported that the sublingual administration of a Poly(I:C)-adjuvanted vaccine appeared safe, as evidenced by the RT-qPCR analysis of inflammatory-related genes utilizing mRNA from peripheral blood white cells (PBWCs) [13]. In PBWCs, the Poly(I:C)-adjuvanted sublingual vaccination upregulated inflammatory genes at 1 day post-vaccination, but the baseline levels were restored at 7 days post-vaccination. The present study investigated the effects of Poly(I:C)-adjuvanted sublingual vaccination on the expression of several inflammatory-related genes, which were *Saa3*, *Tnf*, *IL1b*, *IL6*, *Ifi4*, *Ccl2*, *Timp1*, *C2*, *Aif1*, and *Omp* in mice and *SAA2*, *TNF*, *IL1B*, *IL6*, *CCL2*, *TIMP*, *C2*, *GZMB*, and *AIF1D* in macaques, in the brains (olfactory bulb and pons) of both mice and macaques and compared it with observations in intranasal vaccination. In the olfactory bulbs of mice, the intranasal administration of the Poly(I:C)-adjuvanted vaccine led to the markedly upregulated expression of these genes even 7 days post-vaccination, unlike sublingual vaccination (Figure 2 and Figure 3). The adverse effects that led to upregulated inflammatory genes in the olfactory bulb corresponded to unique molecular events mediated by intranasal vaccination in mice, which resulted from the sensitivity and reactivity of olfactory perception in mice.

In another brain region, the pons, the upregulation of many inflammatory genes (*Saa3*, *Tnf*, *IL1b*, *IL6*, *Ifi4*, *Ccl2*, and *C2*) was still evident 7 days after intranasal vaccination, but not sublingual vaccination, in mice (Figure 3). In the macaque pons, the upregulation of these inflammation-related genes was observed 7 days after intranasal vaccination, but not sublingual vaccination (Figure 4). Intranasal vaccination was reported to induce a high risk of Bell’s palsy with inactivated influenza vaccines [34]. The association between intranasal vaccines and Bell’s palsy was also reviewed in a recent article [35]. Bell’s palsy is most commonly caused by an impairment of the facial nerve that emerges from the facial nerve nucleus in the pons [36]. As mentioned above, vaccine administration via the intranasal route resulted in upregulated inflammation-related genes in the pons 7 days after vaccination in both mice and macaques. The upregulated expression of these genes may induce a proinflammatory condition that impairs the facial nerve. Thus, vaccination-associated Bell’s palsy could be mediated by upregulated gene expression in the pons, which is caused by intranasal vaccines. Based on the gene expression analyses, one should avoid the intranasal administration of vaccines owing to their deleterious effects on the brain, both on the olfactory bulb and pons, and therefore one should look into alternative routes, such as sublingual administration.

### 4.4. Use of PBWCs for Safety Assessment of Vaccines and/or Adjuvants

The lung is a suitable tissue type for the evaluation of the harmful effects of vaccines using gene expression analyses [20]. Except for *C2*, the intranasally administered vaccine significantly upregulated the inflammation-related genes in the mice’s lungs only 1 day after immunization, but not at 1 or 7 days after sublingual vaccination. No upregulation of these inflammation-related genes was observed in the lungs of macaques 7 days after sublingual vaccination. In previous studies, we found that administering a Poly(I:C)-adjuvanted vaccine sublingually upregulated the inflammatory/proinflammatory genes in PBWCs on the first day after vaccination, but the basal levels were restored after 7 days [13]. PBWCs respond to administered vaccinations by upregulating inflammation-related genes in a time course comparable to that of the lungs. These findings imply that PBWCs are a far superior monitoring window than the lungs in assessing the adverse effects of vaccinations and/or adjuvants, as assessing PBWCs is non-invasive, which is advantageous in clinical applications.

Compared to RT-qPCR, histopathological examination was less sensitive in assessing vaccine-induced adverse reactions. The histological method detected few adverse events in the olfactory bulb and pons 7 days after vaccination in macaques, despite the upregulated inflammation-related marker genes in these tissues.

### 4.5. Safety of the Sublingual Poly(I:C)-Adjuvanted Vaccine

An adjuvant is essential for the safety and effectiveness of a mucosal vaccine. Although numerous adjuvants exist, two are particularly notable. MF59 or AS03 is an oil-in-water nano-emulsion that stimulates Th1/Th2 [5]. The other is double-stranded (ds) RNA Poly(I:C), a ligand for TLR3 that induces immunological and proinflammatory responses [6]. MF59 and AS03 have been authorized as adjuvants for intramuscularly administered influenza vaccines. However, the clinical application of Poly(I:C) as a vaccine adjuvant remains unapproved, except for restricted use in oncology. Studies utilizing intranasal vaccination in mice have predominantly indicated the generation of proinflammatory cytokines and associated factors mediated by Poly(I:C) [19,20]. Nevertheless, these proinflammatory reactions were overestimated owing to the hypersensitive reaction observed with the intranasal administration of the Poly(I:C)-adjuvanted vaccination in mice, as mentioned above. In our prior studies utilizing a non-human primate model, the sublingually administered Poly(I:C)-adjuvant vaccine showed safety outcomes equivalent to those of AddaS03, a non-clinical derivative of the AS03 adjuvant, when assessing the total blood count, biochemical blood tests, and plasma CRP. The sublingual Poly(I:C) adjuvant vaccine exhibited gene expression profiles identical to those of AddaS03 concerning proinflammatory cytokines and associated factors (IL12a, IL12b, GZMB, IFN-alpha1, IFN-beta1, and CD69) [13]. The Poly(I:C)-adjuvanted vaccination elicited plasma production levels comparable to those of AddaS03 in inflammation-related cytokines: IFN-alpha, IFN-gamma, IL-12, and IL-17. The safety of the Poly(I:C) adjuvant should be considered comparable to that of AS03 in the case of sublingual vaccinations.

Poly(I:C)-adjuvanted sublingual vaccination evoked hitherto unrecognized immunological responses, including the aberrant upregulation or downregulation of gene expression associated with immune suppression or tolerance, Treg differentiation, and T cell exhaustion [13]. Thus, the Poly(I:C) adjuvant elicits the balanced “Yin/Yang”-like enhancement and suppression of immune responses, rendering it both safe and effective for sublingual vaccination [13,14].

## 5. Conclusions

A Poly(I:C)-adjuvanted influenza HA vaccine administered sublingually had few adverse effects and was assessed as safe based on the RT-qPCR results for inflammatory-associated genes in the brain (olfactory bulb and pons), lungs, tongue, and (submandibular) lymph node in both macaques and mice. Conversely, the intranasally administered vaccination caused deleterious side effects by upregulating these genes in the olfactory bulb, pons, and lungs in both macaques and mice, as well as severe detrimental effects in the brain (olfactory bulb and pons), particularly in mice.

## Figures and Tables

**Figure 1 vaccines-13-00261-f001:**
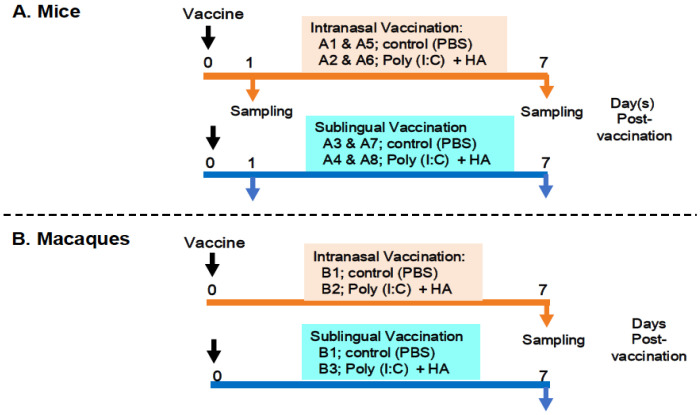
Examination outline for vaccination and sampling. (**A**) shows intranasal and sublingual vaccinations in **mice**—A1 and A5 and A3 and A7 for control (PBS), A2 and A6 and A4 and A8 for vaccines (Poly(I:C+HA)—and sampling time points at 1 day and 7 days post-vaccination. (**B**) shows intranasal and sublingual vaccinations in **macaque monkeys**—B1 for control (PBS), B2 and B3 for vaccines (Poly(I:C+HA)—and sampling time point at 7 days post-vaccination.

**Figure 2 vaccines-13-00261-f002:**
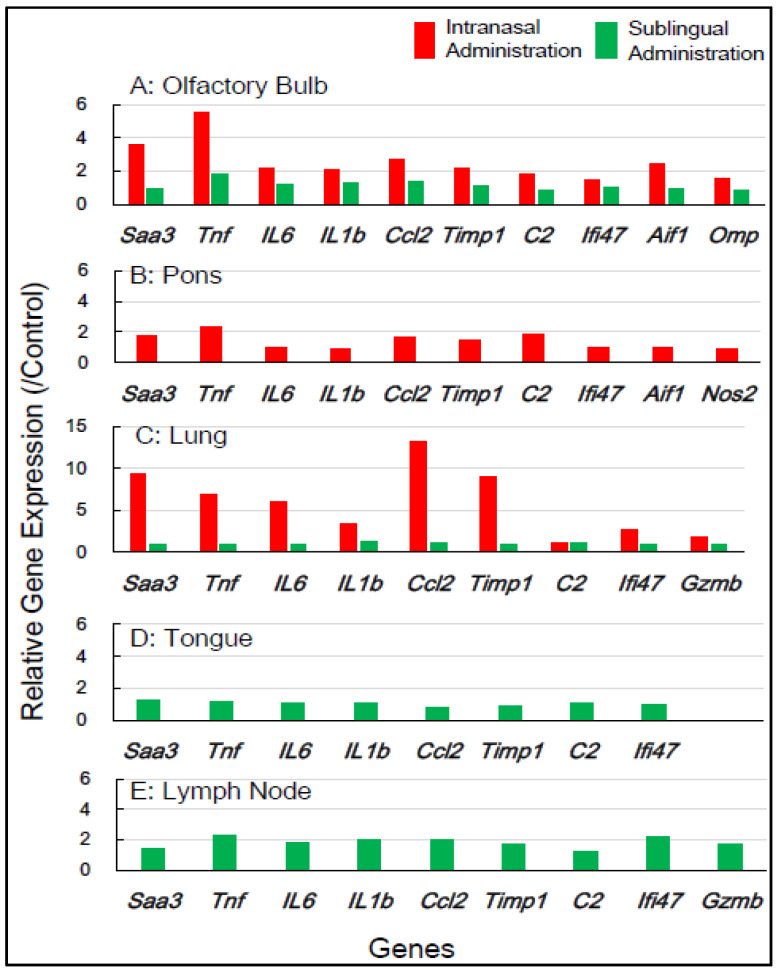
In **mice**, the changes in the expression of inflammation-associated genes in the five tissues/sites at **1 day** post-vaccination via the intranasal and sublingual routes.

**Figure 3 vaccines-13-00261-f003:**
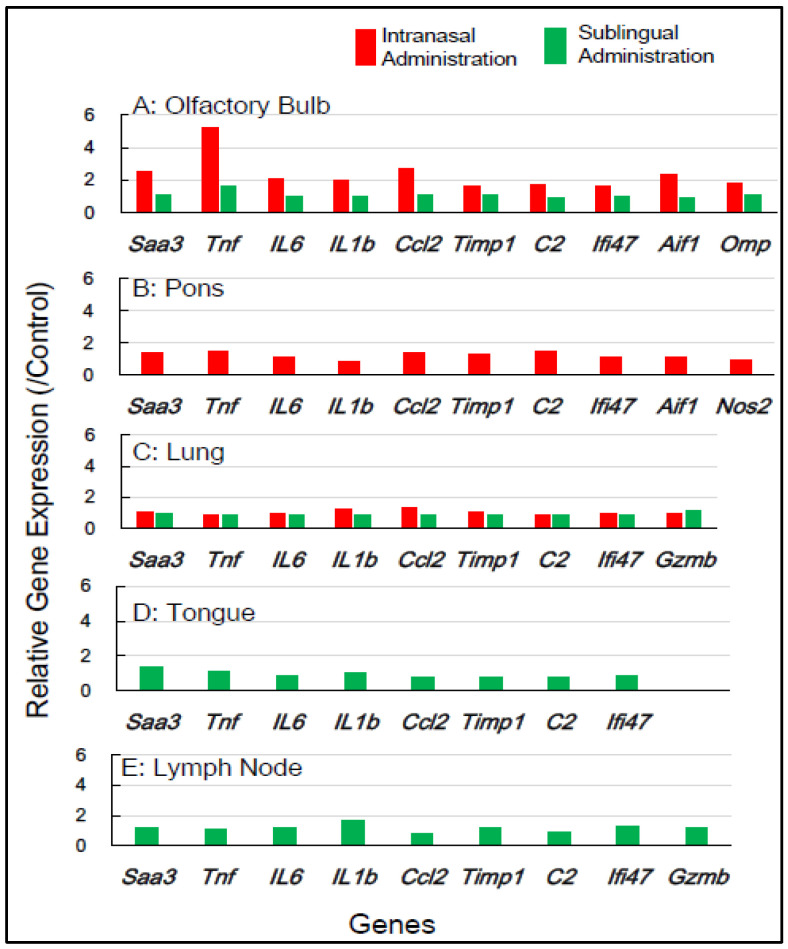
In **mice**, the changes in the expression of inflammation-associated genes in the five tissues/sites at **7 days** post-vaccination via the intranasal and sublingual routes.

**Figure 4 vaccines-13-00261-f004:**
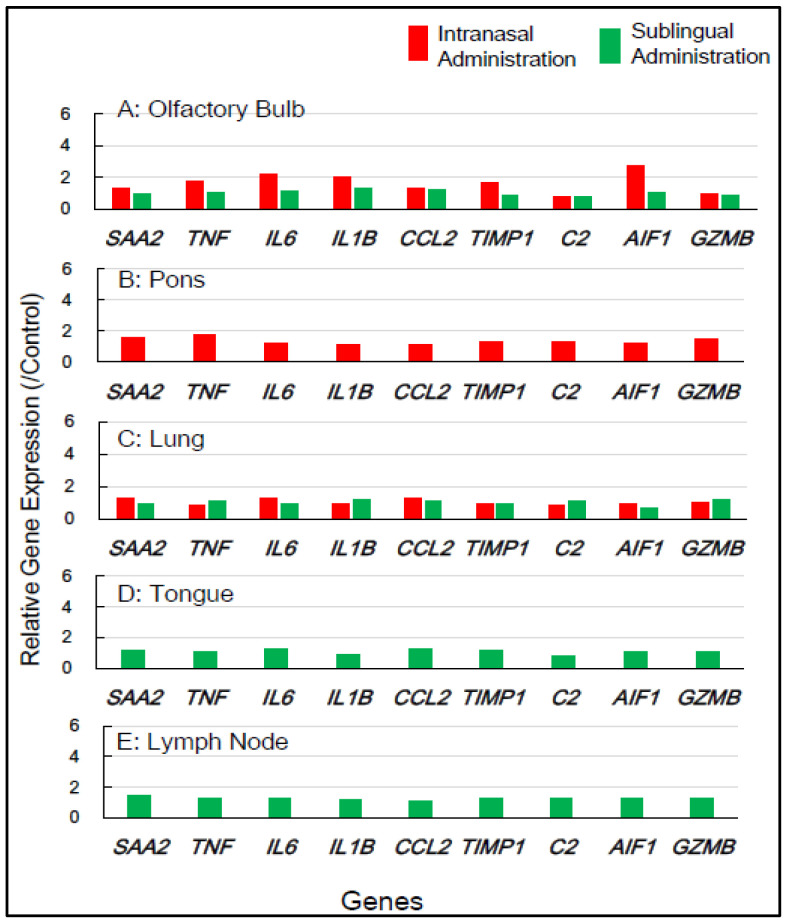
In **macaque monkeys**, the changes in the expression of inflammation-associated genes in the five tissues at **7 days** post-vaccination via the intranasal and sublingual routes.

**Figure 5 vaccines-13-00261-f005:**
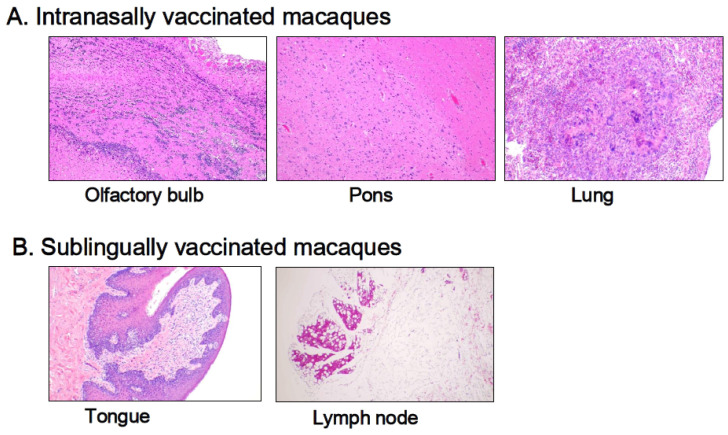
Histopathological observations of HE-stained sections from macaque monkeys intranasally (**A**) and sublingually (**B**) administrated the Poly(I:C)-adjuvanted vaccine.

**Table 1 vaccines-13-00261-t001:** Gene symbols, gene information, tissue/site samples, and time points for gene expression analyses in vaccinated mice.

Gene Symbol	Gene Information Product; Description; Function [Reference]	Sample; Tissue/Site *	Time Point **
** *Saa3* **	Serum amyloid A 3; acute response protein [19,20]	OB, P, L, T, (S) LN	1 d, 7 ds
** *Tnf* **	Tumor necrosis factor; inflammatory cytokine [21]	OB, P, L, T, (S) LN	1 d, 7 ds
** *IL6* **	Interleukin 6; immune–inflammatory response [22]	OB, P, L, T, (S) LN	1 d, 7 ds
** *IL1b* **	Interleukin 1 beta; inflammatory cytokine	OB, P, L, T, (S) LN	1 d, 7 ds
** *Ccl2* **	C-C motif chemokine ligand 2(MCP1); chemokine [23]	OB, P, L, T, (S) LN	1 d, 7 ds
** *Timp1* **	Tissue inhibitor of metalloproteinase 1; tissue repairing protein [19,20]	OB, P, L, T, (S) LN	1 d, 7 ds
** *C2* **	Complement component 2; opsonic function; phagocytic cell activation [20]	OB, P, L, T, (S) LN	1 d, 7 ds
** *Ifi47* **	Interferon gamma inducible protein 47; pathogen defense protein [19,20]	OB, P, L, T, (S) LN	1 d, 7 ds
** *Aif1* **	Allograft inflammatory factor 1; microglial marker [24]	OB, P	1 d, 7 ds
** *Omp* **	Olfactory marker protein; odor detection/signal transduction	OB	1 d, 7 ds
** *Nos2* **	Nitric oxide synthase 2, inducible, iNos; [25,26]	P	1 d, 7 ds
** *Gzmb* **	Granzyme-B; NK cell protease; apoptosis induction [27]	L, (S) LN	1 d, 7 ds

* OB: olfactory bulb, P: pons, L: lung, T: tongue, LN: (submandibular) lymph node. ** (day(s) post-vaccination).

**Table 2 vaccines-13-00261-t002:** Gene symbols, gene information, tissue/site samples, and time points for gene expression analyses in vaccinated macaque monkeys.

Gene Symbol	Gene Information Product; Description; Function [Reference]	Sample; Tissue/Site *	Time Point **
** *SAA2* **	Serum amyloid A 3; acute response protein [19,20]	OB, P, L, T, (S) LN	7 ds
** *TNF* **	Tumor necrosis factor; inflammatory cytokine [21]	OB, P, L, T, (S) LN	7 ds
** *IL6* **	Interleukin 6; immune–inflammatory response [22]	OB, P, L, T, (S) LN	7 ds
** *IL1B* **	Interleukin 1 beta; inflammatory cytokine	OB, P, L, T, (S) LN	7 ds
** *CCL2* **	C-C motif chemokine ligand 2 (MCP1); chemokine [23]	OB, P, L, T, (S) LN	7 ds
** *TIMP* **	Tissue inhibitor of metalloproteinase 1; tissue repairing protein [19,20]	OB, P, L, T, (S) LN	7 ds
** *C2* **	Complement component 2; opsonic function; phagocytic cell activation [20]	OB, P, L, T, (S) LN	7 ds
** *AIF1* **	Allograft inflammatory factor 1; microglial marker [24]	OB, P, L, T, (S) LN	7 ds
** *GZMB* **	Granzyme-B; NK cell protease; apoptosis induction [27]	OB, P, L, T, (S) LN	7 ds

* OB: olfactory bulb, P: pons, L: lung, T: tongue, LN: (submandibular) lymph node. ** (day(s) post-vaccination).

## Data Availability

Data are available from S.N. upon reasonable request.

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
