# Peer review of "Safety Assessment of a Sublingual Vaccine Formulated with Poly(I:C) Adjuvant and Influenza HA Antigen in Mice and Macaque Monkeys: Comparison with Intranasal Vaccine"

_vaccines, 2025, doi:10.3390/vaccines13030261_

Round 1
Reviewer 1 Report
Comments and Suggestions for Authors
Yamamoto et al reported a study investigating the safety of a Poly(I:C) adjuvanted recombinant HA influenza vaccine by blood tests, measuring inflammation-related mRNA expression in several different tissues using RT-PCR and histological examination on mice and macaque monkey models. While the study showed some interesting findings, the manuscript is not ready for publication. I have several major comments as below.
1. Line 48, the description is not correct. Vaccines include many other types besides protein- and nucleic acid- based vaccines, such as live attenuated vaccines and killed vaccines.
2. Line 55, vaccine adjuvants also include many other types. For example, Alum is a very successful adjuvant with a long history of use in vaccines. There are also many other vaccine adjuvants other than emulsions and TLR agonists, such as agonists of STING, RIG-I, NLRs and many combination adjuvant systems. I strongly recommend the authors read some review papers about vaccines and adjuvants, for example, Pulendran, B., S. Arunachalam, P. & O’Hagan, D.T. Emerging concepts in the science of vaccine adjuvants. Nat Rev Drug Discov 20, 454–475 (2021).
3. Line 101, the description is misleading. Clinical trials were not included in this manuscript, so the safety on human should not be concluded without supporting evidence.
4. Line 164, only volume of vaccines was mentioned, how about the amounts of vaccines? The antigen amounts and adjuvant amounts are more related to efficacy and safety, compared with volume. Did the two routes use same amounts?
5. Line 184, what kind of assay was used for plasma samples? Please specify and provide necessary method description.
6. Line 228-248. Data is required for conclusions about “blood tests”.
7. Line 274-275, please provide rationales why day 1 and day 7 were selected as the timepoints for tissue harvests.
8. In figure 2, 3 and 4, how did you calculate the “relative gene expression” shown on the Y-axis? Is it a fold change by PBS group? If it is a fold change, why some genes showed no values in the intranasal group in tongue and lymph nodes because it should be at least around 1 unless they were significantly downregulated. Besides, error bars were missing in the bar graphs. Sample sizes and statistical analysis are also highly recommended to be included in the figure legends.
9. Please add discussions about the limitations of only showing mRNA levels for the study. The functional protein levels are not always consistent with mRNA levels.
10. Please provide the full name of several abbreviations, such as “PBWCs”.
11. There are some typos and confusions in the sentences, such as in line 284, “internasal” should be intranasal. In line 416-417, should it be “adverse effects…have been described from intranasal vaccines?”
Comments on the Quality of English LanguageThe language accuracy needs to be improved.
Author Response
"Please see the attachment

Reviewer 2 Report
Comments and Suggestions for Authors
Yamamoto et al present a manuscript entitled 'Safety Assessment of a Sublingual Vaccine Formulated with Poly(I:C) Adjuvant and Influenza HA Antigen in Mice and Macaque Monkeys: Comparison with Intranasal Vaccine'. The authors examine the safety of an Influenza subunit vaccine consisting of HA and polyI:C, a well-known TLR3 agonist, in male mice and macaques. Mice and macaques were vaccinated with a single dose comparing intranasal versus sublingual application. A group of genes associated with inflammation was examined at day 1 and 7 postvaccination in mice and day 7 post-vaccination in macaques in different tissues, and used as an indicator of vaccine safety. Intranasal vaccination in mice raised expression of Saa3 and TNF in olfactory bulb, pons, and lung on day 1. In the case of olfactory bulb, mRNA levels were enhanced at day 7 as well. Lung mRNA levels from intranasally vaccinated mice also indicated increased levels of inflammation whereas lungs from sublingually vaccinated mice did not. In macaques, there was some upregulation of inflammatory genes in olfactory bulb and pons, but differences between intranasal and sublingual vaccination were minimal. No histopathological differences were noted in the lungs of intransally versus sublingually infected macaques. The data in general suggests that intranasal application leads to higher increases in inflammatory gene expression than sublingual vaccination. Especially upregulation in the olfactory bulb and pons are noted as safety concerns. However, interpretation of the results is clouded by the acknowledgement of higher sensitivity of the mouse olfactory system to foreign substances, and some weaknesses in the methods (below). This reduces the impact of the paper.
Major concerns
The main and only assay, gene expression (Figures 2-4), has weaknesses. The assay needs additional details as to how the increases in expression were calculated, and what exactly the y-axis units mean. Does a ratio of 1 mean there is no increase? There are no statistics to gauge which differences are significant.
The main outcome appears more complex than the general conclusions state. For example, sublingual administration enhances inflammatory gene expression in the lymph node whereas intranasal does not; sublingual administration also raises inflammatory gene expression (albeit less than sublingual); expression increases after intranasal application appear minor in pons for both mice and macaques. In addition, the mouse olfactory system is more sensitive to foreign reagents and this may not translate into humans. Finally, what level of inflammation is necessary to induce a protective response while not causing excessive inflammation? To equate low levels of inflammatory markers with safety is somewhat simplistic in the absence of additional data such as protection from challenge. Inflammation in the lung may be needed to induce a strong adaptive response. The data in general may suggest that sublingual administration of a protein vaccine with adjuvant is safer than a intranasal one. But in absence of statistics and any differences in tissue pathology, and the higher sensitivity of the mouse olfactory system, any conclusion is not straightforward. Some of these issues are not adequately discussed.
Figure 5. The conclusion here is no detrimental effects. However, which parameters were examined in Fig 5? Was only immune cell infiltration measured, as stated in the results? Also, it should be compared to a unvaccinated animal.
Minor concerns
The authors use techniques to reduced the mucin barrier and excessive dilution of the samples upon vaccination. It should be included in the results section that these measures were taken, and it would be helpful to discuss what the outcome was of these aids to optimal vaccination.
The amount of HA and polyI:C should be stated in materials and methods, not just refer to a previous paper.
Reviewer 3 Report
Comments and Suggestions for Authors
A comparative study of intranasal and sublingual vaccination with the Poly (I:C)-adjuvanted influenza HA vaccine was conducted in mice and macaque monkey models. Blood and tissue samples were collected at 1 day and 7 days post-vaccination. The expression levels of the inflammation-associated genes were determined using RT-qPCR. It has been shown that intranasally administered Poly(I:C)-adjuvanted vaccination significantly upregulated the expression of inflammatory-related genes, while sublingually administered vaccine did not produce any adverse effect. A drawback of the study is that the immune response to vaccination was not investigated. This raises the question whether the complete tolerance of sublingual vaccination could be associated with a low immune response.
Note
1) Line 275. Figure 2 shows the change inexpression of inflammation-associated genes in the five tissues 1 day post-vaccination with the intranasal vaccine
Data on intranasal vaccination are provided for only three tissues.
2) Line 284 …the internasal vaccine on...
Technical error.
Round 2
Reviewer 1 Report
Comments and Suggestions for Authors
The authors answered my questions and I dont' have any more concerns.
Comments on the Quality of English LanguageI don't have more comments.